# Elements of a stochastic 3D prediction engine in larval zebrafish prey capture

Andrew D Bolton[1†*], Martin Haesemeyer[1], Josua Jordi[1], Ulrich Schaechtle[2], Feras A Saad[2], Vikash K Mansinghka[2], Joshua B Tenenbaum[2], Florian Engert[1]

[1]Center for Brain Science, Harvard University, Cambridge, United States; [2]Brain and Cognitive Sciences, Massachusetts Institute of Technology, Cambridge, United States

**Abstract** The computational principles underlying predictive capabilities in animals are poorly understood. Here, we wondered whether predictive models mediating prey capture could be reduced to a simple set of sensorimotor rules performed by a primitive organism. For this task, we chose the larval zebrafish, a tractable vertebrate that pursues and captures swimming microbes. Using a novel naturalistic 3D setup, we show that the zebrafish combines position and velocity perception to construct a future positional estimate of its prey, indicating an ability to project trajectories forward in time. Importantly, the stochasticity in the fish's sensorimotor transformations provides a considerable advantage over equivalent noise-free strategies. This surprising result coalesces with recent findings that illustrate the benefits of biological stochasticity to adaptive behavior. In sum, our study reveals that zebrafish are equipped with a recursive prey capture algorithm, built up from simple stochastic rules, that embodies an implicit predictive model of the world.

*For correspondence:
andrewdbolton@fas.harvard.edu

Present address: †Harvard University, Cambridge, United States

Competing interests: The authors declare that no competing interests exist.

## Introduction

It is becoming clear from recent ethological and neuroscience studies that remarkable capabilities in animals are often built by combining sets of more basic behaviors. For example, seemingly complicated behaviors like schooling in fish arise from stereotypic visuomotor rules being executed by members of the group, without reference to the emerging global pattern (*Couzin and Krause, 2003*). Bees learning to pull strings for hidden rewards appear to be displaying insight and ingenuity, but are in fact instituting a set of observational and associative learning rules in sequence (*Alem et al., 2016*). The synthesis of intelligence from the interaction of 'mindless' behavioral modules has long been a staple of computer science and artificial intelligence (*Brooks, 1991*; *Minsky, 1988*; *Braitenberg, 1986*). However, a precise mechanistic explanation of how intelligent behavior is generated has remained elusive.

Intelligence itself can been defined as the practice of model-building about ourselves and our surroundings (*Lake et al., 2017*). Indeed, humans and animals have evolved internal models that allow us to both predict ongoing dynamics in the environment and anticipate how our own actions give rise to consequences in the world (*Battaglia et al., 2013*; *Ullman et al., 2017*; *Baillargeon, 1987*; *Mischiati et al., 2015*; *Borghuis and Leonardo, 2015*). Much progress has been made on the neurobiological and computational principles that could mediate these complex abilities, but how exactly internal models are built from component behavioral parts is unknown. Moreover, many theories of mind rely on deterministic digital logic, while the brain generates intelligence using noisy, stochastic units in neurons (e.g. *McCulloch and Pitts, 1943*). Noise in *any* computing system is usually considered inconvenient and a nuisance to be overcome (*Körding and Wolpert, 2004*). However, there is precedent for noisy sensory detection and stochastic movements working to the benefit of many animals. Crayfish and paddlefish, for instance, both take advantage of stochastic resonance to detect

sparse prey and predators (*Douglass et al., 1993*; *Russett et al., 1999*). Beneficially stochastic foraging has been observed in animals ranging down to micro-organisms, while predated animals use mixed stochastic strategies to avoid predictability by predators (*Jensen, 2018*).

Can internal models that mediate predictive abilities be reduced to a set of simple rules that are benefited by the stochasticity of neural systems? We posit that explicit physical intelligence is likely built upon a framework of more primitive sensorimotor behaviors that constitute an implicit model of how objects exist and move within the world (i.e. *Brooks, 1991*). Therefore, characterizing the goals, algorithms, and advantages of animals possessing implicit models should provide insight into the evolution of more advanced forms of predictive knowledge (*Spelke and Hespos, 2018*).

Here, we examine these questions through characterization and computational modeling of 3D prey capture sequences executed by the larval zebrafish. The larval zebrafish, a teleost with ~100,000 neurons, is a tractable model organism amenable to an array of modern neuroscience techniques (*Avella et al., 2012*; *Dunn et al., 2016*). Prey capture requires the zebrafish to select, pursue, and ultimately consume fast moving single-celled organisms swimming through its environment. We chose this model system and behavior for a multitude of reasons. Foremost, we hypothesized that the pursuit of fast-moving prey should be benefitted by internal models; the ability to extrapolate prey trajectories forward in time and the prediction of how each pursuit movement impacts prey position would both, a priori, appear to be helpful in capturing fast-moving objects (e.g. *Yoo et al., 2019*; *Borghuis and Leonardo, 2015*). Furthermore, there is precedent for the zebrafish constructing relatively complex behaviors from simple rules. For example, stabilization of position in turbulent streams is accomplished by instantiating a curl-detector for local water flow and an optomotor response (*Oteiza et al., 2017*; *Naumann et al., 2016*), while energy-efficient foraging emerges from the zebrafish's innate tendency to locomote via alternating series of unidirectional turns (*Dunn et al., 2016*). Finally, the zebrafish's ongoing behavior is largely probabilistic, reflecting the stochasticity of its neural systems. For instance, the precise number of unidirectional turns in any spontaneous swimming stretch is stochastic, while turn magnitude in response to angular optic flow varies widely (*Dunn et al., 2016*; *Naumann et al., 2016*).

We build upon foundational studies in zebrafish prey capture (*Patterson et al., 2013*; *Trivedi and Bollmann, 2013*; *Bianco et al., 2011*) by analyzing this behavior in its natural 3D setting, whereas previous studies have typically neglected vertical fish and prey movements. Moreover, we develop an experimental and computational framework that can simultaneously record fish and prey trajectories. This approach allowed us to accurately map the fish's sensorimotor transformations in response to ongoing prey features, which are described in an egocentric spherical coordinate system that specifies the fish's three-dimensional point of view. Particularly, we illustrate three main elements of the fish's prey capture algorithm that reflect an implicit intuitive model of physics. First, sensorimotor transformations during prey capture are largely controlled by the azimuth angle, altitude angle and computed radial distance of prey before the fish initiates a pursuit movement. Second, all aspects of the fish's 3D movement choices are strongly and proportionally modulated by the angular and radial velocity of its prey. Combining these two rules yields an emergent strategy whereby the fish predicts future prey locations and recursively halves the angle of attack. Third, we show that the speed of the fish's recursive hunting strategy is benefited by noise in its sensorimotor transformations. Importantly, this stochasticity is graded, meaning that the further away a prey item is from the fish's goal, the more variable the outcome of the fish's choice becomes. Using a series of computational models and virtual prey capture simulations, we show that position perception, velocity projection, and graded variance are *all* essential for effective and energy-efficient prey capture performance. This suggests that the fish's implicit models are, in fact, adaptive to the animal.

Overall, this work reveals that even the most complex behavior in larval zebrafish can be reduced to a set of simple rules. These rules coalesce to generate a stochastic recursive algorithm embodied by zebrafish during hunting, which ultimately reflects an implicit predictive model of the world.

## Results

### Developing a 3D environment for elucidating hunting sequences in zebrafish

We first sought to characterize the sensorimotor transformations larval zebrafish implement when pursuing and capturing paramecia. Hunting sequences were recorded from 46 larval zebrafish using a behavioral setup that could simultaneously image the fish and its prey from the top and side at high resolution and speed (*Figure 1A*). Custom computer vision software was designed to reconstruct the fish's 3D position and two of the principal axes, yaw and pitch (*Figure 1B*), as well as the trajectories of paramecia in the environment (*Figure 1C*). These reconstructions allowed us to spatially map prey position and velocity to an egocentric spherical coordinate system originating at the mouth of the fish. Hereby, each paramecium is assigned an azimuth, altitude and distance as a positional coordinate ('Prey Az, Prey Alt, Prey Dist'), along with angular (Prey δAz / δt, Prey δAlt / δt) and radial (Prey δDist / δt) velocities with respect to the fish's 3D point of view (*Figure 1D*). To view the prey environment from the fish's reconstructed 3D perspective, see *Video 1*.

Larval zebrafish swim in discrete 'bouts', which consist of a pulse of velocity lasting ~200 ms, followed by a variable period of intermittent quiescence (*Figure 2A*; *Budick and O'Malley, 2000*). We took advantage of this unique facet of fish behavior to frame each bout performed during a hunting sequence as an individual 'decision' based on the spherical position and velocity of pursued prey. Specifically, we identified the start-time and end-time of each swim bout using fluctuations in tail variance and velocity (see Materials and methods). This allowed us to precisely understand how the fish transforms pre-bout prey features into movements that displace and rotate the fish to a new location in 3D space at the end of the bout.

Hunting sequences themselves consist of an initiation bout, multiple pursuit bouts, and a termination bout. Initiation bouts were identified by detecting whether the eyes on a given bout have converged. Eye convergence, which allows the use of stereovision by creating a small binocular zone, is a well-known correlate of hunting state entrance in zebrafish (*Gahtan et al., 2005*; *Bianco et al., 2011*). Hunt sequences were therefore identified by clustering the continuous eye angle record for both eyes during each bout (*Figure 1B*, *Figure 1—figure supplement 1A*). Hunt sequences were terminated on the bouts where fish either struck at their pursued prey or clearly quit pursuit. Quitting has been called an 'aborted' hunt in the literature (*Henriques et al., 2019*; *Johnson et al., 2019*), and most aborts in our dataset, as in other studies, corresponded to the cluster demarking deconvergence of the eyes (*Figure 1—figure supplement 1B*) and a return to an exploratory state. The average hunting sequence ending in a strike in our dataset lasted for five bouts (Interquartile Range = 4 to 7).

### Zebrafish typically choose the closest prey item when initiating a hunt sequence

Nearly all hunt sequences in our dataset began with the choice of a single prey item to pursue (*Figure 1—figure supplement 1B*). The choice of prey item was straightforward. Fish almost invariably chose the closest paramecium in the environment conditioned on the fact that the paramecium was fairly close to its midline in azimuth and significantly above it in altitude (*Figure 1—figure supplement 2*; $\mu_{az} = 0.7°$ $\sigma_{az} = 48.4°$; $\mu_{alt} = 19.9°$ $\sigma_{alt} = 19.7°$; $\mu_{dist} = 3.4$ mm $\sigma_{dist} = 1.6$ mm). There was no particular bias of prey choice in terms of direction or magnitude of velocity (*Figure 1—figure supplement 2A*, bottom panels).

### Sensorimotor transformations during prey capture are largely controlled by Pre-Bout prey position

After choosing a prey item during an initiation bout, the fish engages in a series of pursuit bouts (see *Video 1*) that can each influence the position, yaw, and pitch of the animal (*Figure 2A*). Pursuit bouts are conducted until prey are positioned in a 'strike zone', which defines the termination condition for successful hunts in spherical coordinates relative to the fish: this zone is directly in front of, and considerably above the fish (*Figure 2B*; avg. 0.9° Prey Az, 17.4° Prey Alt, .870 mm Prey Dist; *Mearns et al., 2019*). We investigated whether displacements and rotations during pursuit bouts were influenced by 3D prey position before each bout. For the rest of this manuscript, only

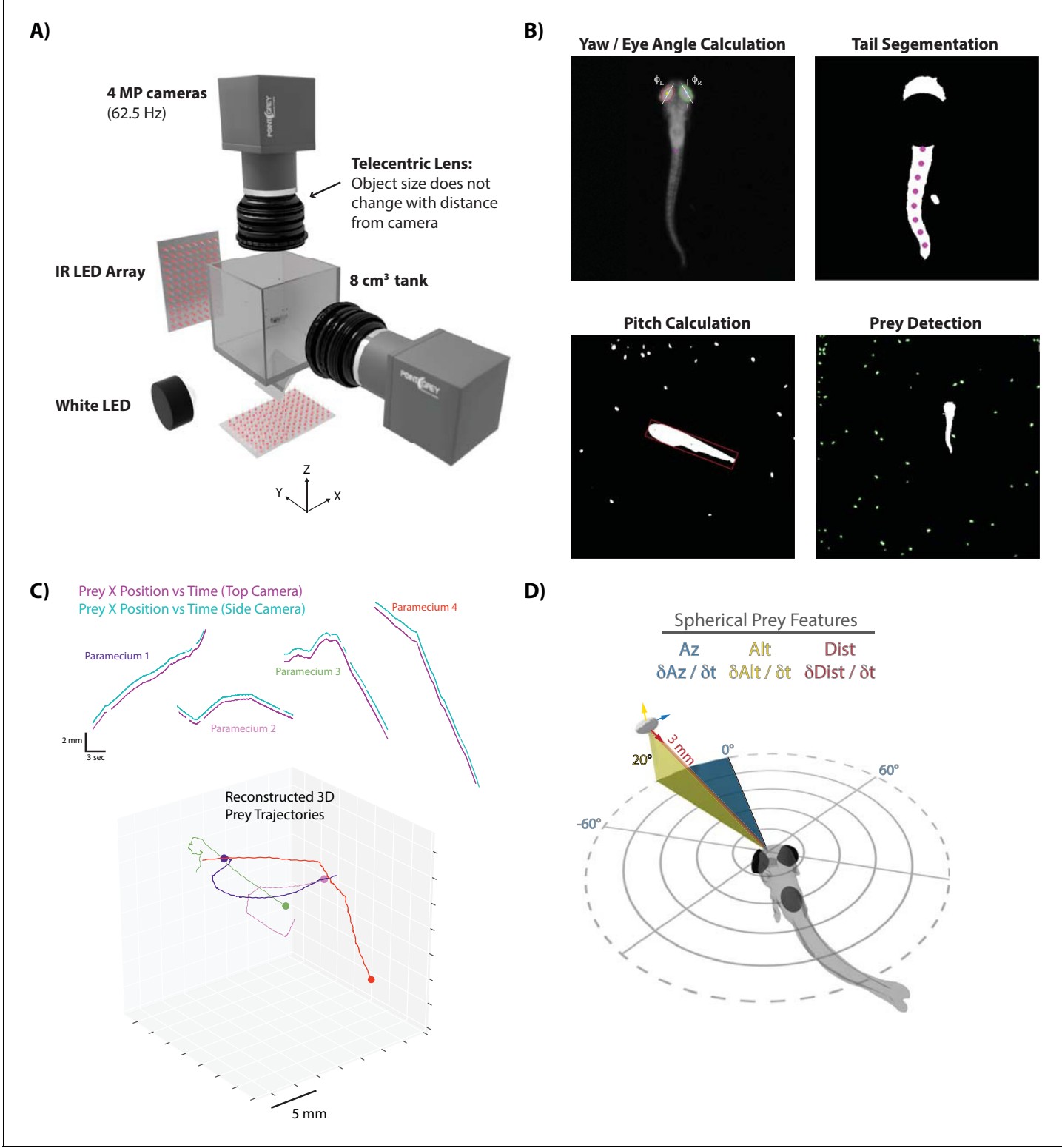

**Figure 1.** A novel 3D experimental paradigm for mapping prey trajectories to fish movement choices. (**A**) 3D rendering of rig design and features. (**B**) Computer vision algorithms extract the continuous eye angle, yaw, pitch, and tail angle of the zebrafish. In every frame, prey are detected using a contour finding algorithm. (**C**) Prey contours from the two cameras are matched in time using a correlation and 3D distance-based algorithm, allowing 3D reconstruction of prey trajectories. (**D**) Prey features are mapped to a spherical coordinate system originating at the fish's mouth. Altitude is positive above the fish, negative below. Azimuth is positive right of the fish, negative left. Distance is the magnitude of the vector pointing from the fish's mouth to the prey.

*Figure 1 continued on next page*

*Figure 1 continued*

The online version of this article includes the following source data and figure supplement(s) for figure 1:

**Figure supplement 1.** Identificaiton of hunt sequences.
**Figure supplement 2.** Fish tend to choose the closest available prey when initiating hunt sequences.
**Figure supplement 2—source data 1.** Source data describing prey at hunt initiations.

sensorimotor transformations during hunt sequences in which a strike was performed are described. However, the algorithm the fish uses during aborts is nearly identical (*Figure 2—figure supplement 1*; note that the last three bouts in abort sequences tend to go awry for unknown reasons; *Henriques et al., 2019*).

For every pursuit bout, we calculate an axis of motion in egocentric spherical coordinates along which fish displace during the bout. This axis is defined by an azimuth and an altitude angle ('Bout Az', 'Bout Alt'), and the magnitude of displacement along this axis is termed 'Bout Distance'. Because the axis of motion during bouts is not perfectly aligned to the axis of symmetry, yaw and pitch changes are independently described per bout ('Bout ΔYaw', 'Bout ΔPitch'). Diagrams of rotation and displacement variables are provided alongside *Figure 2C and D*.

Each bout aspect is primarily controlled by the position of the prey relative to the fish immediately preceding bout initiation (*Figure 2C and D*). Regression fits show that Bout Az and Bout ΔYaw are well correlated to Prey Az, with negligible offset (0.3° and 0.48°). This transformation simply implies that fish displace and turn towards their prey. A similar linear relationship is seen when mapping Bout Alt and Bout ΔPitch of pursuit bouts to Prey Alt, but this time with significant negative offsets (−15.13° and −1.79°). These negative offsets imply that if Prey Alt before a pursuit bout is 0°, which one may preconceive as the fish's ultimate 'goal', the fish will dive downwards by ~15° and rotate downward by ~2°, thereby consistently maintaining the prey above itself at the end of pursuit bouts. Schematics of these rules are shown in *Figure 2E and F*. Interestingly, these positional transformations reflect the fish's preferred position in which to strike for prey consumption: with prey directly in front and significantly above the fish (*Figure 2B*).

Bout Distance along the axis of motion established by Bout Az and Bout Alt is a more complex variable and will be addressed below.

## All 3D movements during prey capture are strongly modulated by prey velocity

Fish must capture prey that can move very quickly (in our assay, 74% 3D vector velocity >3 paramecium lengths per second, 27% > 6 paramecium lengths per second; avg. Prey $\delta Az / \delta t$ = 29°/s, Prey $\delta Alt / \delta t$ = 25°/s; *Figure 3—figure supplement 1A*). We therefore surmised that prey velocity perception should be required for prey capture. Previous studies had suggested that kinematics of zebrafish movements change from slow to fast bouts based on whether prey are approaching or swimming away from the fish (*Patterson et al., 2013*). Our 3D setup allowed us to conduct a detailed analysis of prey velocity perception in all planes.

We find that every movement the fish performs during prey capture is strongly and proportionally influenced by both the radial and angular velocity of its prey (*Figure 3*). All angular bout variables are amplified if prey are moving away from the fish (*Figure 3* light colors) and dampened when prey are moving towards the fish (*Figure 3* dark colors; *Figure 3—figure supplement 1B*). Modulation of bout features is *proportional* to the velocity of the prey, as it is well

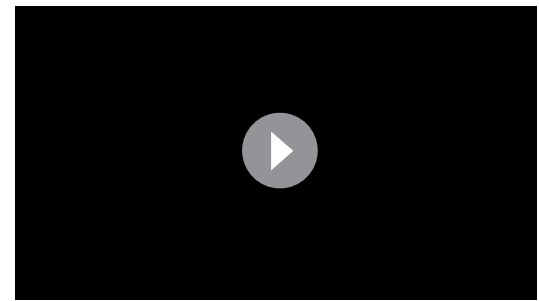

**Video 1.** In each instance, a hunt is shown from the top and side cameras simultaneously, followed by a virtual reality reconstruction of the fish's point of view during the hunt. The virtual reality reconstruction, built in Panda3D, is generated from 3D prey coordinates and unit vectors derived from the 3D position, pitch, and yaw of the fish.
https://elifesciences.org/articles/51975#video1

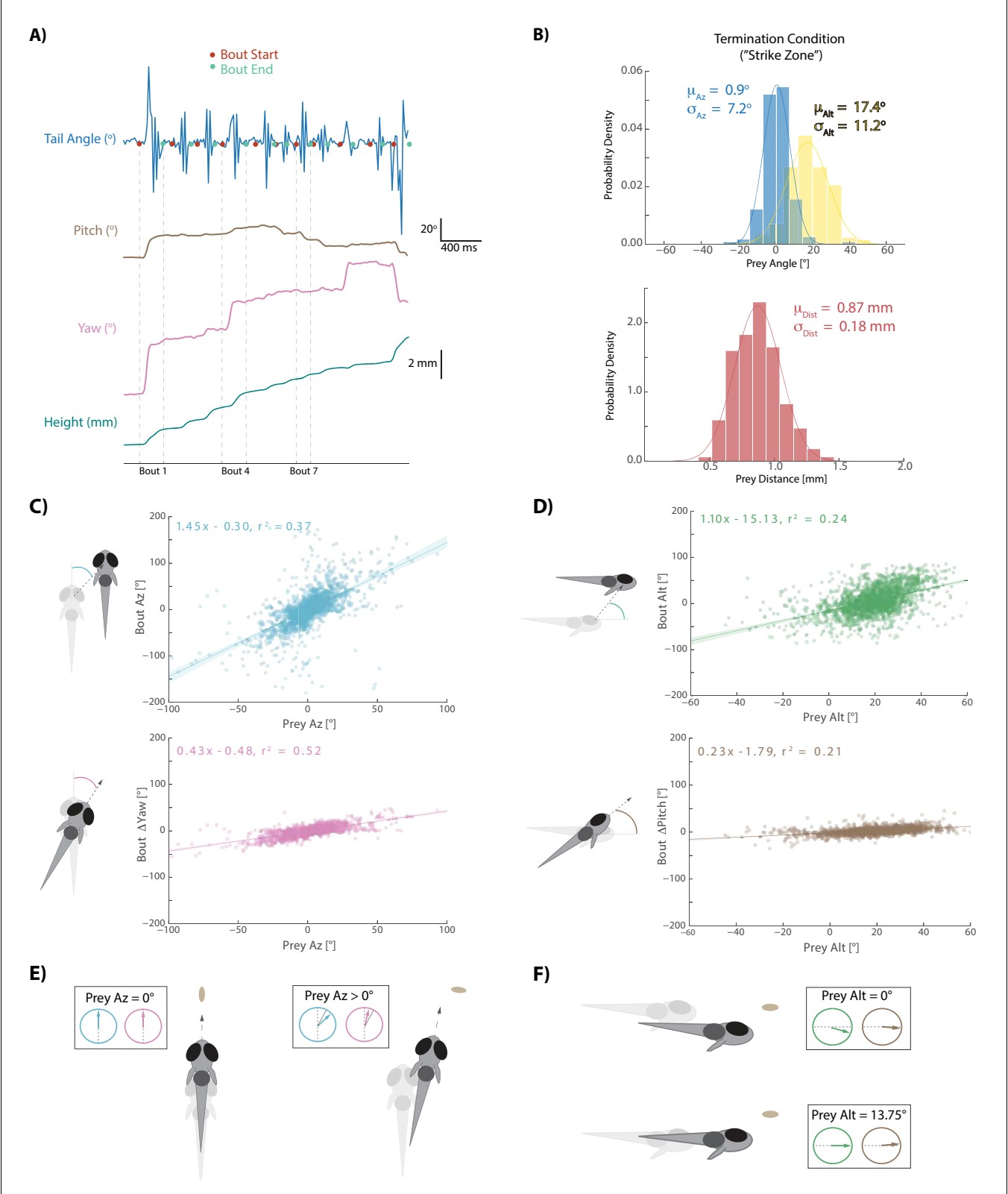

**Figure 2.** Fish execute 3D movements based on the position of their prey. (A) During hunt sequences, fish swim in bouts that can be detected using tail variance. Bouts can change the yaw, pitch, and position of the fish, while time between bouts is marked by quiescence. (B) Histograms showing the distributions of spherical prey positions when fish successfully ate a paramecium during a strike. (C, D) Regression fits between prey position and bout variables executed by the fish. (E, F) Features of sensorimotor transformations based on prey position: fish swim forward if prey are directly in front.

*Figure 2 continued on next page*

*Figure 2 continued*

Otherwise, if prey are on right, fish displace and rotate right; and vice versa. Fish displace downward if prey are at 0° altitude, but displace with no altitude change if prey are at 13.75°. In all schematics (**C–F**), positions and orientations at the beginning of the bout are represented by transparent fish, and by opaque fish at the end of the bout.

The online version of this article includes the following source data and figure supplement(s) for figure 2:

**Source data 1.** Source data for all bouts conducted in dataset.

**Figure supplement 1.** Prey capture algorithm during aborted hunt sequences.

fit by multiple linear regression transforming position *and* velocity of prey into fish bout variables (see *Figure 3* legend).

We next wondered whether velocity perception constitutes a prediction of future prey position. Given that most bouts zebrafish make during hunts are 176 ms long (IQR = 144 ms to 208 ms), we can calculate from each prey velocity the change in prey position that would occur during an average bout. We find that zebrafish transform **projected** prey position changes by 1.43 for Bout Az,. 31 for Bout ΔYaw, 1.70 for Bout Alt, and. 18 for Bout ΔPitch (see *Figure 3* legend). Critically, these values closely approximate the coefficients describing bout transformations to prey position itself in *Figure 2*. This is the first hint that the fish has reduced the problem of position prediction to adding velocity multiplied by bout time to its current prey position percept. In this sense, the fish are performing Euler's Method of approximating a future position based on its instantaneous derivative.

The final bout variable to address, Bout Distance, is more nuanced than the other four bout features. The linear relationship between Bout Distance and Prey Distance, in general, is only strong when prey are <4 mm from the fish (*Figure 3C*). Bout Distance is significantly modulated by radial prey velocity ($\delta$Dist / $\delta$t; see *Figure 1D*) when prey are within 2 mm (*Figure 3C*). In this spatial window, radial velocity of prey coming toward the fish dampens Bout Distance while radial velocity moving away from the fish amplifies Bout Distance. Indeed, multiple regression finds an overall correlation between Bout Distance and Prey Distance with significant dampening when Prey $\delta$Dist / $\delta$t < 0 and amplification of Bout Distance when Prey $\delta$Dist / $\delta$t > 0 (*Figure 3C* legend for coefficients).

## Computational models of prey capture behavior show efficiency and success arise from velocity perception

Next, we asked to which degree the use of prey velocity contributes to the animal's ability to efficiently and successfully capture prey. To that end we constructed a Virtual Prey Capture Simulation Environment in which computational models of fish behavior with different 'powers' can be pitted against each other. Each of these models varied along two axes: how the fish perceives its prey and how it transforms prey perception into movements. The relative powers possessed by each model allowed us to assess how the fish balances prey capture speed and accuracy with the energy required for such a movement sequence. We started by re-creating paramecium trajectories of 225 hunt sequences initiated by the fish that resulted in a real-life strike (*Figure 4A*, Materials and methods). From the initial conditions of these trajectories, we launched five models that transform current paramecium features into 3D bouts as the prey moves through the environment.

Each model initiates a bout at the precise moment when the real fish initiated a bout during the sequence, and model sequences are terminated when the prey enters the virtual strike zone (*Figure 2B*, Materials and methods). Moreover, we compare all models to the real 3D fish trajectory (the 'real fish' model, *Figure 4* Model 1 [Blue]), both assuring that our 3D bout and strike-zone characterizations recapitulate real-life performance, and allowing us to compare post-bout coordinates of all models to the characterization of real bouts.

The capabilities of each model are as follows (*Figure 4A*): First, a multiple regression model was fit on only positional features of the paramecium (Model 2 [Orange]). This model linearly transforms current position of the prey into 3D bout features according to the regression fits in *Figure 2C,D* (and the bout distance fit from 3C). Model 3 (Green), a multiple regression model fit on both the position and velocity of prey, accounts for the amplification and dampening of bout features by prey

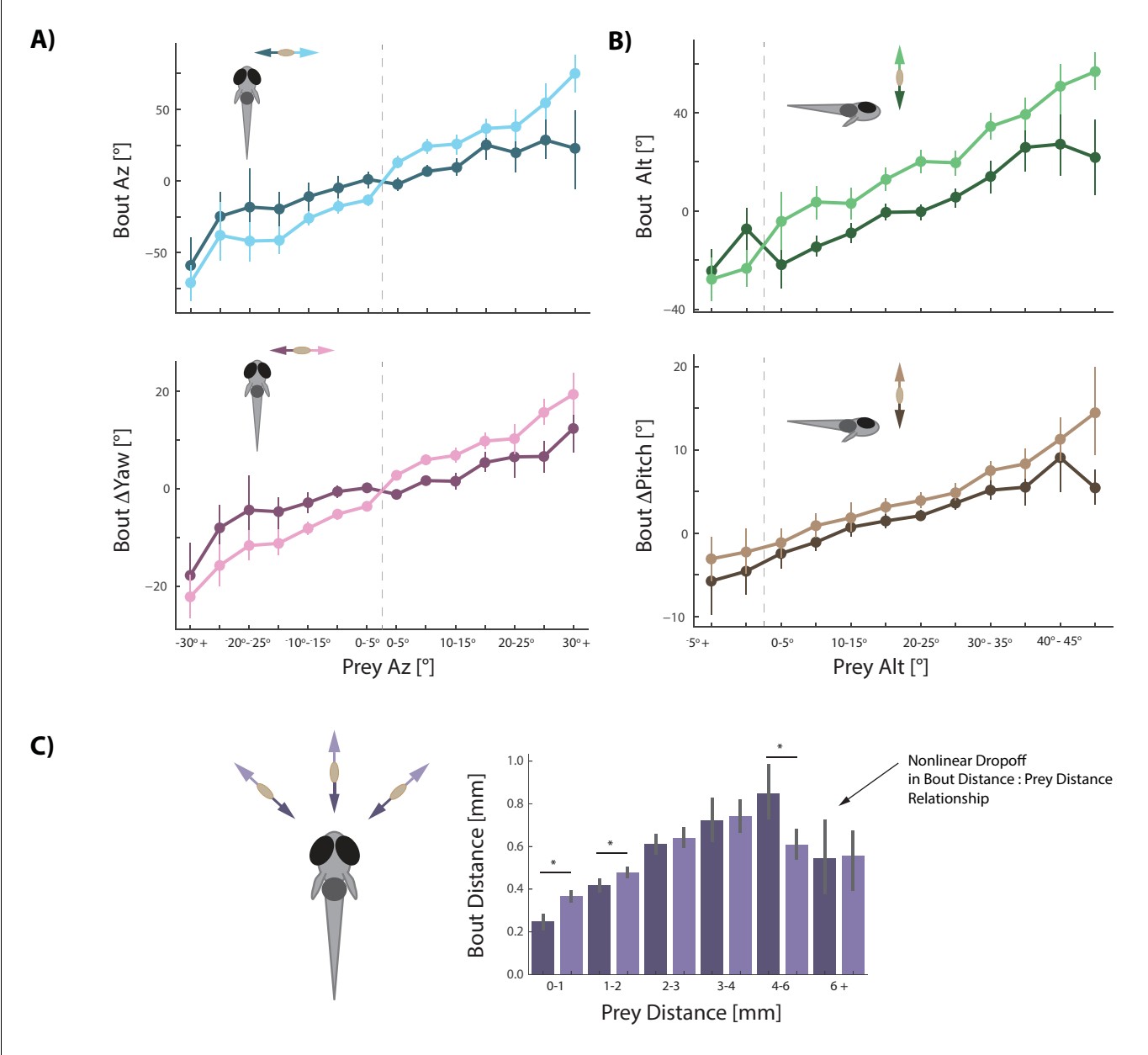

**Figure 3.** Prey velocity biases all aspects of fish's bout choices. (A, B) Light colors depict each bout variable if prey are moving away from the fish while dark colors indicate that prey are moving toward the fish in the same 5° window of space. A pattern emerges of dampening movements when prey velocity is towards the fish, and amplifying movements when prey velocity is away from the fish. Point locations are means with error bars representing 95% CIs. Multiple regression fitting of bout variables to prey position and velocity in all planes confirm and quantify the dampening and amplification (azimuth velocity moving left to right of the fish is positive, altitude velocity upward is positive). Bout Az is biased by .251 * Prey δAz / δt (.219 - .283 95% CI), Bout ΔYaw by .054 * Prey δAz / δt (.046-.061 95% CI), Bout Alt by .300 * Prey δAlt / δt (.268 - .331 95% CI), and Bout ΔPitch by .031 * Prey δAlt / δt (.024 - .039 95% CI). Dividing these coefficients by the average bout length (0.176 s) yields the projected positional coefficients described in the text. (C) Bout Distance is linearly proportional to Prey Distance but only within 4 mm of the fish, with breakdown in relationship above 6 mm. Likewise, dampening of Bout Distance when Prey δDist / δt < 0 (prey approaching radially), and amplifying when Prey δDist / δt > 0 (prey moving afar), occurs in two windows: 0–1 mm, and 1–2 mm from the fish. Overall, multiple regression finds: *Bout Distance = 0.105 * Prey Dist + .053 * Prey δDist / δt* (95% CIs = 0.094-.116, .034-.071), which reflects the fact that ~70% of pursuit bouts occur when prey are within 2 mm. (*: p<0.05/6, Bonferroni corrected two-tailed t-tests. p-values: 0–1 mm = 2.7 * $10^{-6}$, 1–2 mm: 0.0016, 2–3 mm: 0.38, 3–4 mm: 0.73, 4–6 mm: .00046, 6+: 0.92).

The online version of this article includes the following figure supplement(s) for figure 3:

**Figure supplement 1.** Distribution of prey velocity and example prey velocity-based biasing of bout features.

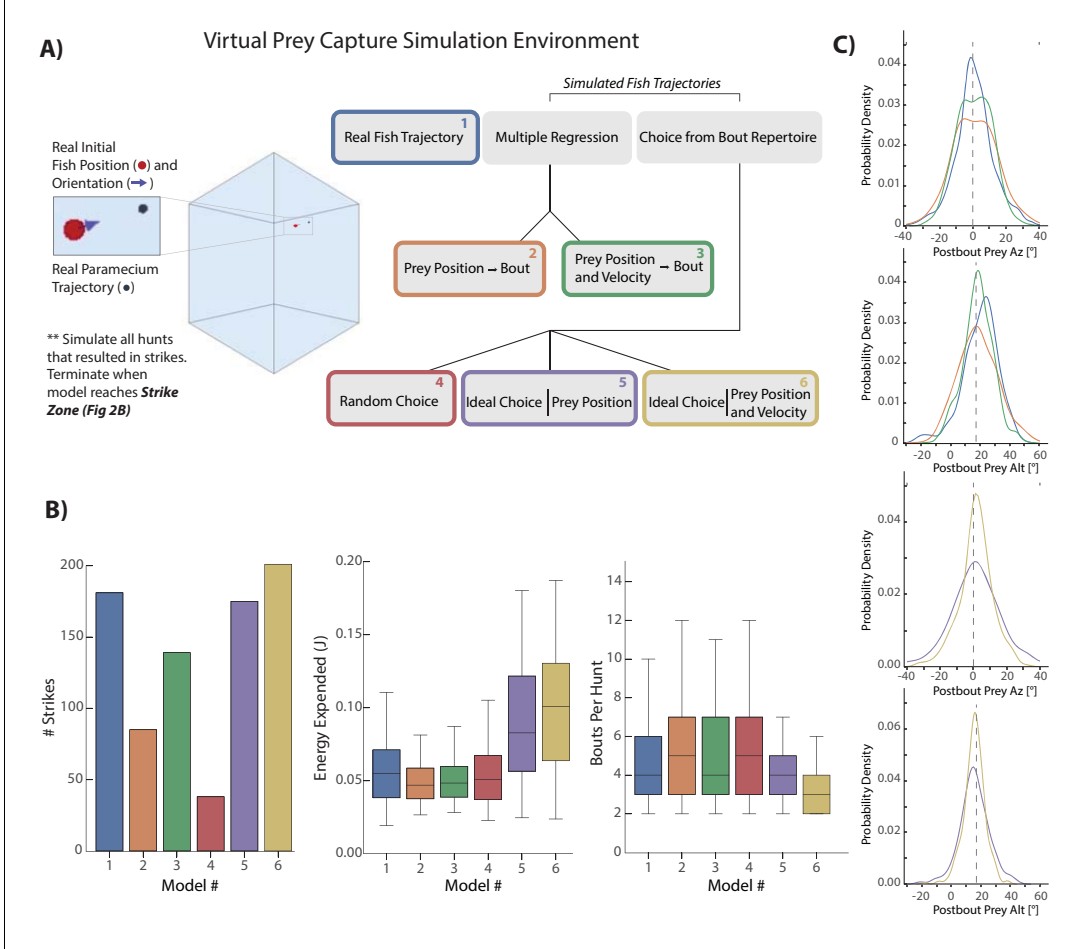

**Figure 4.** Virtual prey capture simulation reveals necessity of velocity perception. (**A**) A virtual prey capture environment mimicking the prey capture tank was generated to test three different types of models, and six models overall, described in the schematic. Models control a virtual fish consisting of a 3D position (red dot) and a 3D unit vector pointing in the direction of the fish's heading. Virtual fish are started at the exact position and rotation where fish initiated hunts in the dataset. Prey trajectories are launched that reconstruct the real paramecium movements that occurred during hunts. The virtual fish moves in bouts timed to real life bouts, and if the prey enters the strike zone (defined by the distributions in *Figure 2B*, Materials and methods), the hunt is terminated. (**B**) Barplots (total #) and box plots (median and quartiles) showing performance of all six models in success (# Strikes), energy use per hunt sequence, and how many bouts each model performed during the hunt (a metric of hunt speed). (**C**) KDE plots showing the distribution of Post-Bout Prey Az and Post-Bout Prey Alt distributions for each model during virtual hunts. Dotted lines demark the strike zone mean.

velocity in *Figure 3*; position transforms are thus linearly biased by the velocity coefficients described in the *Figure 3* legend.

Models 4–6 are 'Choice' models which can draw from a distribution of 1782 pursuit bouts conducted by fish in our dataset during sequences ending in a strike. This bout pool can be thought of as the 'pursuit repertoire' of larval zebrafish. Model 4 (Red) simply assembles random bouts drawn from the bout pool into a hunt sequence. Model 5 (Purple) chooses the ideal bout from the pool, with each bout scored on its achievement in reducing the prey's azimuth, altitude, and distance to the mean values of the strike zone (see Materials and methods). Model 5 does *not* have access to the velocity of the prey, meaning that it will zero in on the *pre-bout* prey position. Lastly, Model 6 (Gold) has the same capabilities to choose ideally as Model 5, but will extrapolate the current prey velocity and add its time multiplied bias to the current position. Therefore, Model 6 can predict the future paramecium position at the end of the bout, but chooses ideally instead of linearly.

To compare the models, we describe four facets of their performance intended to score raw hunting success as well as energetic cost: First, how many times out of 225 the model achieved success in placing the virtual prey in its strike zone. Second, how much total energy was expended by the

bout combination used during the hunt (see Materials and methods). Third, how many total bout choices the model made per hunt as a metric for capture speed (*Figure 4B*). And lastly, the Post-Bout Prey Az and Alt prey coordinate for each transformation was plotted to illustrate how well each model does in reducing prey coordinates to the strike zone (*Figure 4C*). Performance of each model for an example prey trajectory can be viewed in *Video 2*.

The first clear result is that the velocity-based regression Model 3 (Green) improves hunting success over the position-only regression Model 2 (Orange) by 65%. Moreover, the average number of bouts is one less for the velocity model, matching the average of the real fish (Blue). This indicates that the velocity information processed by the fish, which allows projection of future prey coordinates, is critical for both its success rate and speed in capture. When examining the Post-Bout Prey Az for the regression models, the velocity-based Model three shows a tighter distribution around 0°, indicating that it is closer to the strike zone on average than position-only Model 2 (*Figure 4C*, right top panels). This is also true for Post-Bout Prey Alt, where the green plot shows a stronger bias toward the strike zone.

Velocity information is also critical to the performance of ideal choice models. As expected, the random choice Model 4 (Red) performs very poorly, indicating that although similar 'types' of pursuit bouts are chosen, success can only be gained by accounting for prey features. Model 5 (Purple), interestingly, does not outperform the real fish in terms of success (3% worse) or average speed of capture, and expends significantly more energy. Model 5 therefore issues high energy bouts (i.e. bouts that strongly rotate and displace the fish), but without any average improvement over what the fish actually did, owing to the high velocity of the average prey item (*Figure 3—figure supplement 1A*). Model 6 (Gold), however, by accounting for prey velocity and choosing ideally, improves success rate over Model 1 (Real Fish) and 5 (Ideal Position) by 14% and 17%, reduces the average number of required bouts by 1, and more effectively reduces Post-Bout Prey Az and Alt to the strike-zone (*Figure 4C*, bottom panels). Nevertheless, Model 6 expends the most energy of any model per hunt sequence, meaning that fully ideal choice comes at an energetic cost.

We therefore conclude that position perception improves performance over issuing random pursuit bouts with no reference to the prey, and that velocity information in all formats improves model performance over position perception alone. Of note, the high energy usage of the ideal models relative to the real fish argues against the natural implementations of these seemingly optimal strategies. Lastly, although the real fish takes fewer bouts to reach the target than the regression models (#2, #3), it requires slightly more total energy to do so. This implies that a modicum of additional energy is expended per bout, and we speculate that the generation of stochasticity in the real fish's algorithm (described below) is to blame.

## Pre-Bout to Post-Bout prey coordinate transformation reveals a canonical hunting strategy

We next wondered which of the numerous prey capture strategies observed in nature arises from the rules implemented in *Figures 2* and *3*. For example, a strategy of immediately zeroing the angle of attack is used by many predators and is called 'pure pursuit'. Other common strategies include 'deviated pursuit', where a constant angle of attack is maintained throughout the hunting sequence. We find that larval zebrafish use neither of the two, but rather reduce Prey Az and Prey Alt by a factor of 0.5 at each bout. In the first two panels of *Figure 5A* (blue and yellow), the post bout angle of attack is plotted against the pre bout angle, and a constant ratio of ~0.5, regardless of prey velocity direction, is apparent in the slopes of the regression lines. This strategy of reducing prey angle in both planes to a fixed proportion on each bout is

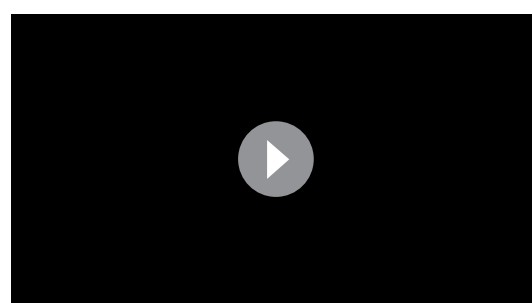

**Video 2.** Virtual prey capture simulation environment. Each model from *Figure 4* begins at the same position and orientation and is given the task of hunting the same paramecium trajectory. In this representative hunt sequence, every model except the Random Choice and Multiple Regression (Position Only) models consume the prey (indicated by red STRIKE flash). As is typical in the simulations, the Ideal Choice (Position) model lags the Ideal Choice (Velocity) model by one bout.
https://elifesciences.org/articles/51975#video2

schematically depicted in *Figure 5B* and *Figure 5—figure supplement 1*. Notably, when prey are below the fish, the slope changes to 0.9, meaning that the fish is only getting 10% closer in altitude per bout. This poor performance for negative Prey Alt is consistent with our observation that hunt initiations are triggered almost exclusively by prey located above the fish. In fact, 92.4% of all bouts in our dataset occur when prey are above 0° Alt (*Figure 5—figure supplement 2*).

Next, we analyzed how much the radial distance to the prey object is reduced as a consequence of each pursuit bout. We find that on average fish become 16% closer to the prey per bout, or in other words, the Prey Distance is scaled by 0.84 at every iteration. This reduction, however, is significantly less for prey objects that move radially away from the fish (0.87) versus when prey move toward it (0.81; *Figure 5A*, light red vs dark red regression fits), which is consistent with the fact that the fish does not modulate Bout Distance based on radial prey velocity when prey are further than 2 mm away (see *Figure 3C*). However, once prey are maneuvered to within 2 mm, radial velocity is taken into consideration and the two fits converge on similar outcomes. Importantly, 69% of pursuit bouts occur when prey are 0–2 mm from the fish, indicating that Bout Distance is typically modulated by radial prey velocity, and that fish can most often achieve a preferred Post-Bout Prey Distance after a bout is completed.

To summarize, because fish account for prey velocity in all directions (*Figure 3*), they are on average capable of achieving a fixed proportional reduction of Prey Az, Alt, and Dist during pursuit bouts. These proportions are consistent from the beginning to the end of hunting sequences (*Figure 5C*); therefore, this data reflects a 'deterministic recursive algorithm' for prey capture: The fish recursively transforms current prey coordinates into more favorable prey coordinates by a fixed scale factor in all planes until the strike zone is attained (*Figure 5D*).

## Graded stochasticity in Pre-Bout to Post-Bout prey coordinates benefits hunting efficiency

Interestingly, we noticed a clear graded increase in variance of the post-bout coordinate in all spherical planes as pre-bout coordinates trended away from the strike zone (*Figure 5A*, *Figure 6A* left panels, *Figure 6—figure supplement 1*). This suggested to us that the fish is implementing a stochastic recursive algorithm rather than the deterministic recursion described in *Figure 5D*.

In order to capture the stochasticity of pre-bout to post-bout prey transforms made by zebrafish during pursuit, we chose to use probabilistic generative models (Dirichlet Process Mixture Models: 'DPMMs'). These models are Bayesian, non-parametric models which avoid the key problem in the statistical modeling field of having to arbitrarily specify the number of variables that best characterizes your data (*Gershman and Blei, 2012*). They can be thought of as 'probabilistic programs' that accurately mimic fish choices given a particular pre-bout prey coordinate (*Goodman et al., 2008*; *Mansinghka et al., 2015*; *Cusumano-Towner et al., 2019*). Using this framework, we sought to uncover whether a stochastically implemented version of the recursive hunting algorithm was beneficial to the fish. To that end, we pitted a stochastic algorithm defined by our Bayesian model against the deterministic algorithm (5D) and compared performances of both models to each other and to the performance of the real fish (*Figure 6B*). Remarkably, the stochastic model outperformed the deterministic model in terms of capture speed (# Bouts to Capture, *Figure 6B*; Wilcoxon signed rank = $7.5 * 10^{-17}$). This suggested that the fish's strategy of graded stochastic transforms centered on a preferred post-bout value is actually beneficial versus accurately achieving a fixed, preferred post-bout value. Moreover, although everything except initial prey position has been abstracted away, the performance of the stochastic model approaches that of the real fish (*Figure 6B*, right panel).

To confirm that graded stochasticity in our Bayesian model was responsible for its increased prey capture speed, we simply injected proportional noise into deterministic bout choices and asked whether we could speed up prey capture (*Figure 6C,D*). This was clearly the case (Wilcoxon Signed Rank: $1.87 * 10^{-9}$ azimuth, $3.96 * 10^{-14}$ distance); common scenarios that we observed in these simulations are illustrated in *Figure 6C*. The deterministic algorithm definitively achieves the strike zone in a fixed number of transforms (*Figure 6C*, top), but injecting proportional noise can either directly improve on deterministic choices ('Beneficial' panel, 6C), start poorly but then recover and outpace deterministic choices ('Recovery', 6C), or perform detrimentally ('Detrimental', 6C). Nevertheless, the average performance when injecting proportional noise is typically equivalent or better by one bout (*Figure 6D*). Given that successful hunts in our dataset had an interquartile range of 4 to 7 total

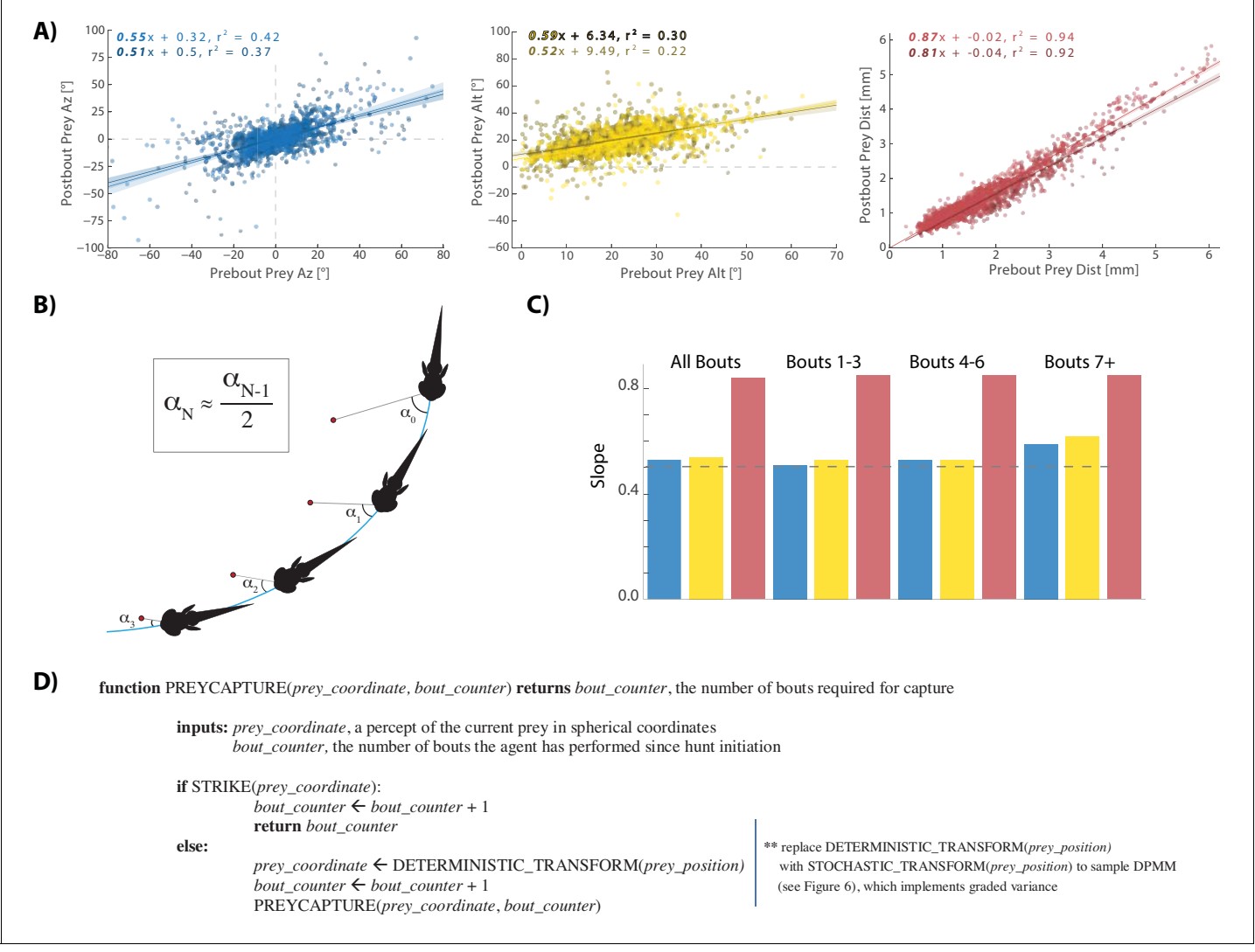

**Figure 5.** A cannonical 'halving' strategy emerges. (A) Regression plots showing relationships between pre-bout prey coordinates and post-bout prey coordinates. Dark colors, prey are moving toward the fish. Light colors, prey are moving away from the fish. 95% CI on azimuth transforms' y-intercept includes 0˚. Top right panel fit on all distances < 6 mm (see *Figure 3C*). (B) Schematic showing recursive halving of the angle of attack during pursuit. (C) Regression slopes are constant across the hunt sequence; color coded to 5A. (D) Pseudocode describing the recursive prey capture algorithm that transforms according to 5A until it arrives at the strike zone. The combined (both velocity directions) distance transform is *.84 * Pre-Bout Prey Dist -. 0125 mm = Post Bout Prey Dist*. The azimuth transform is *.53 * Pre-Bout Prey Az = Post Bout Prey Az*. The altitude transformation is *.54 * Pre-Bout Prey Alt + 8.34˚ = Post Bout Prey Alt*. Implementing these equations recursively will terminate the algorithm at 18.1˚ Prey Alt (since .54 * 18.1˚ + 8.34˚ = 18.1˚) and 0˚ Prey Az (.53 * 0˚ = 0˚), which aligns precisely with the strike zone described in *Figure 2B*. See Appendix for full pseudocode of all sub-functions. The online version of this article includes the following figure supplement(s) for figure 5:

**Figure supplement 1.** Schematic showing that the fish will reduce the post-bout angle of attack to the same value regardless of whether prey is moving towards or away from the fish (see *Figure 5A*).

**Figure supplement 2.** Regression fits between Pre-Bout and Post-Bout Prey Alt differ depending on whether prey altitude is positive or negative before the bout.

**Figure supplement 3.** Transformation by the *initiation* bout of Pre-Bout to Post-Bout Prey Az and Alt.

bouts, improving by one bout constitutes an average 14–25% gain in capture speed. We therefore conclude that the fish's graded stochasticity produces performance that is curiously beneficial to the fish while hunting prey.

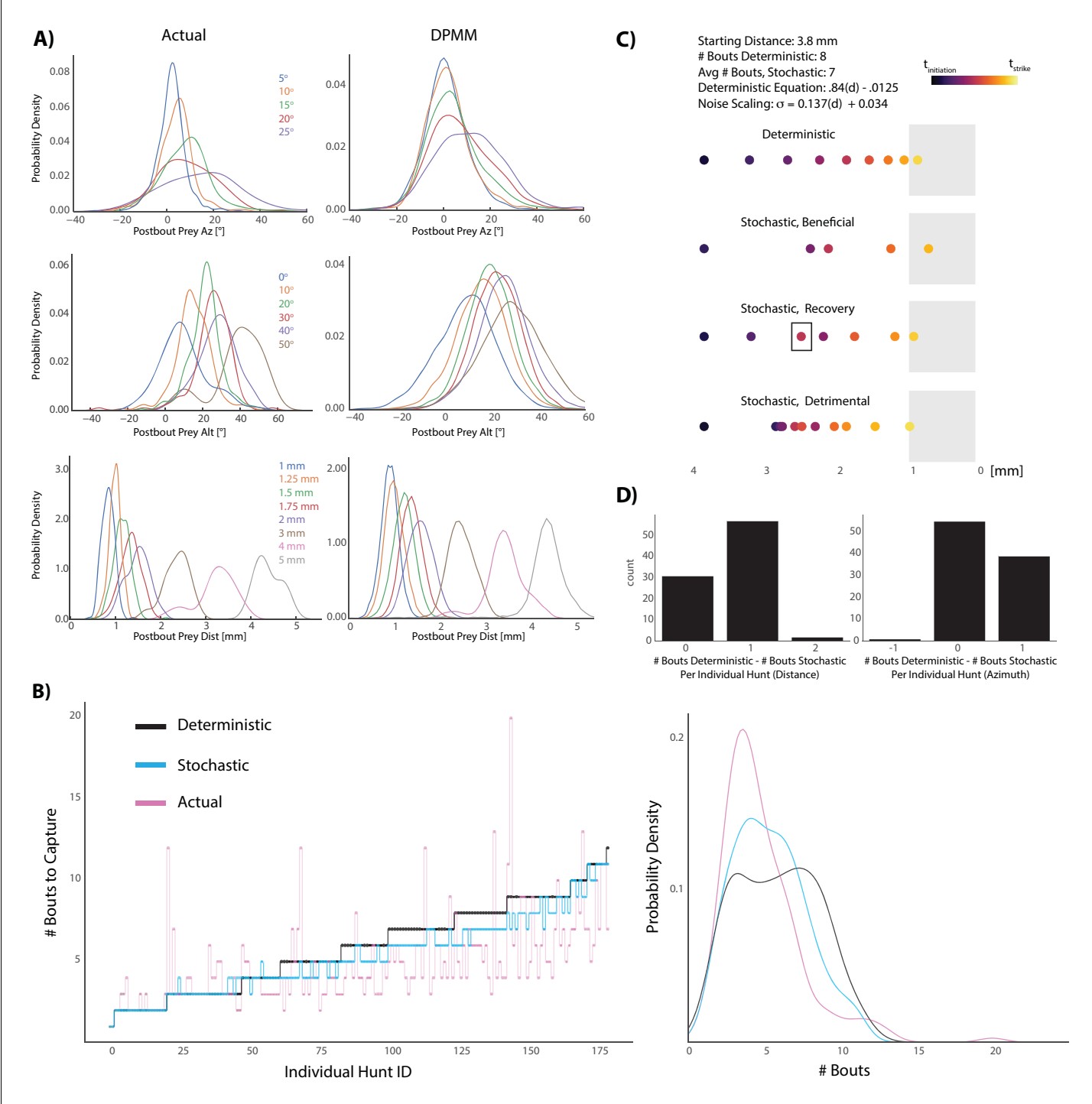

**Figure 6.** Graded stochasticity in sensorimotor transformations improves hunting performance. (A) KDE plots of post-bout variable distributions for pre-bout input coordinates described in the legend, color-coded to the KDEs (i.e. the blue KDE in Az is the distribution of all Post-Bout Prey Az given that Pre-Bout Prey Az is 5°). Real data is binned in 5° windows for angles and. 25 mm bins for distance. DPMM-generated post-bout variables are directly simulated from the model 5000 times, conditioned on the pre-bout value in the legend. (B) The median performance of the DPMMs embedded in a recursive loop (stochastic recursion algorithm; run 200 times per initial prey position) typically ties or outperforms the deterministic recursion model, which transforms with the same pre-bout to post-bout slopes as the DPMMs. Pink line is the performance of the real fish. Right panel: KDE plot of data from 6A, showing that the stochastic algorithm approaches the speed of the real fish. (C, D) A Graded Variance Algorithm where proportional noise is injected into each choice (equations below) is applied 500 times per initial distance (0.1 mm to 10 mm, .1 mm steps) or azimuth (10° to 200° in steps of 2°). Termination condition is a window from. 1 mm to 1 mm for distance, −10° to 10° for azimuth (see Appendix for full algorithm). Deterministic

*Figure 6 continued*

recursion algorithm (*Figure 5D*) is also run on each initial azimuth and distance with .53 * az for azimuth *and .84 * dist - .0125 mm* as the fixed transforms. Graded Variance uses these exact transforms as the mean while injecting graded noise: $\sigma_d = 0.137 * dist + 0.034\ mm$; $\sigma_{az} = .36 * az + 7.62°$, which were fit using linear regression on samples generated by our Bayesian model (6B). (C) shows examples of Graded Variance performance ('stochastic') vs. deterministic performance for an example start distance of 3.8 mm. (D) is a barplot comparing the deterministic performance for each input distance and azimuth to the median Graded Variance ('stochastic') performance where we directly injected noise. Average performance using noise injection typically ties or defeats deterministic choices by one bout.

The online version of this article includes the following source data and figure supplement(s) for figure 6:

**Source data 1.** Generators for BayesDB simulations.

**Figure supplement 1.** Standard deviation is plotted for each individual fish per five degree bin of prey space, indicating that graded stochasticity is not simply observed in the pooled bout population, but at the level of single fish.

## Discussion

In our study, we uncovered three basic rules that larval zebrafish implement while hunting their fast-moving prey: 1) prey position linearly governs the aspects of five degrees of freedom in which fish can rotate or translate through the water: rotation in yaw and pitch, as well as lateral, vertical and radial displacement; 2) prey velocity modulates all of these aspects of 3D motion and allows the fish to project prey position forward in time; and 3) prey coordinate transformation operates via graded variance based on prey proximity to the strike zone. The first two rules are interesting, since their 3D features and specific contingencies have not been examined in zebrafish, and it certainly was not clear that position prediction was already implemented at the larval stage. This general strategy of using velocity to increase prey capture efficiency, however, is implemented in many hunting organisms such as dragon flies and salamanders (e.g. *Borghuis and Leonardo, 2015*), while superposition of position and motion information has been observed in *Drosophila* (*Bahl et al., 2013*). The third rule, which describes the benefits of stochasticity in these hunting algorithms has, to our knowledge, not been described in any trajectory prediction assay to date.

As we show, the aforesaid rules work in tandem to generate excellent, energy efficient performance in prey capture (*Figures 4* and *6*), which is the most complex behavior that larval zebrafish perform and would appear to require elements of physical knowledge. We focus our discussion on how these rules apply to related studies on the neural mechanisms of prey capture, how examining prey capture at two levels of abstraction was beneficial to our study, and how the fish's algorithms appear to be built around the inherent constraints of its own body.

### From algorithms to neurobiology

With respect to the neural implementations of the algorithms we describe, the transformation of angular prey *position* into informative neural activity is encapsulated to a large extent by the abundant research related to retinotopic maps (e.g. *Sperry, 1963*; *Apter, 1946*; *Muto et al., 2013*; but see *Avitan et al., 2016*). The encoding of distance to an object has been well studied in mammalian visual neuroscience and is primarily focused on binocular disparity allowing stereoscopic comparison of each eye's retinotopic map (*DeAngelis et al., 1998*), with some studies focusing on monocular motion parallax (e.g. *Nadler et al., 2008*). Neuronal encoding of object speed, however, requires more sophisticated circuitry and much less is known about its implementation during prey capture. Speed sensitive neurons that discriminate between 'fast' and 'slow' prey-like stimuli have been uncovered in the zebrafish optic tectum (*Bianco and Engert, 2015*), while 'small field' tectal neurons that respond to velocity have been found in toads (*Ewert, 1987*).

Unlike these and other reports studying velocity during prey capture (*Trivedi and Bollmann, 2013*; *Patterson et al., 2013*; *Monroy and Nishikawa, 2011*), we specifically contend that velocity perception is used to point-estimate a future prey position, and that the fish conducts bouts to achieve this estimate, on average, by biasing their prey position-controlled movements. We show that the perception and projection of velocity is key to prey capture success, and that without it, the azimuth and altitude coordinates of prey after bouts are less likely to lie near the strike zone (*Figure 4*). This type of predictive use of velocity is reminiscent of elegant behavioral studies that have illustrated trajectory prediction in salamanders and dragonflies (*Mansinghka et al., 2015*; *Borghuis and Leonardo, 2015*). Quantitative descriptions of such complex algorithms are an

absolute necessity for generating hypotheses about neural implementations (*Marr, 1982*), and virtual prey capture setups for head fixed larvae provide promising inroads for testing our work at the neural level (*Bianco and Engert, 2015*; *Avitan et al., 2016*; *Trivedi and Bollmann, 2013*). Relatedly, monkeys also predict future locations of virtual prey using Newtonian physical attributes in a very similar paradigm to that shown here. In the monkey brain, these attributes are all reflected by neural activity in the dorsal anterior cingulate, which has no known homolog in the zebrafish, nor in other simple animals that use predictive prey models (*Yoo et al., 2019*). This suggests that in more primitive organisms the necessary computations are executed in earlier evolved brain areas which also might play an essential role in the primate.

The stochasticity that gives rise to the graded variance we describe can have several biological sources. It will be important to unravel whether the neural command signals arriving at muscles grow in variance with increased amplitude or if instead the muscles receive fixed input from the brain for a given prey condition, but themselves respond with graded noise. If the source of the variance is largely of a neuronal nature, then it would be interesting to study where exactly in the pathway from sensory areas to motor neurons such noise starts to appear (*Stern et al., 2017*). Further elucidation of the fish's probabilistic strategy should eventually integrate our findings into more general theories describing the utility of noisy biological behavior (see *Jensen, 2018*; *Wiesenfeld and Moss, 1995*, for review) and intentionally probabilistic circuit structures for solving computational problems (*Mansinghka et al., 2008*).

## Strategic behavior arises from simple behavioral rules

The stochastic recursive algorithm in *Figure 6* describes the progression of pre-bout to post-bout prey coordinates without explicitly accounting for prey velocity or specifically executing fish movements. This transformation pattern, which reveals a preferred future prey position and thus a trajectory prediction ability, emerges from the execution of the position and velocity-based rules described in *Figures 2* and *3*. Zebrafish prey capture in *Figure 6C* has, in fact, been reduced to a single input and a recursive series of stochastic divisions with a termination condition, which largely recapitulates the performance of the fish (*Figure 6B*, DPMM). It is unlikely that we would have found such a straightforward description of proportionality and stochasticity at the lower level of abstraction (i.e. in the fish's actual sensorimotor transformations), because pre-bout to post-bout prey transformation is a formulation of fish movements along five different axes acting simultaneously. Describing prey capture in this way allowed us to assess the goals of the fish on each bout (i.e. *Marr, 1982*), revealed that the fish possess an implicit model of how objects move in the world, and may lead to descriptions about how fish are evaluating their own performance during prey capture.

With regard to the benefits of the zebrafish's strategy, the proportional reduction of angle and distance saves energy at the expense of speed (*Figure 4*). Ideal bout choice improved speed of prey capture in our modeling data (*Figure 4*, Model 6); but it did so at the price of spending almost twice as much energy per paramecium captured. This suggests that the increase in feeding rate that the ideal model would afford seems not to be essential for providing an adaptive advantage. Interestingly, the inherent stochasticity in the algorithm significantly improves the speed of capture (*Figure 6*) while adding only a modicum of energy expenditure (*Figure 4*: comparison of Real Fish Model 1 vs. deterministic regression Model 3). This suggests that the fish has evolved a proper balance between energy expenditure and speed of capture. On the whole, the evolution of an efficient algorithm for prey capture in the zebrafish is in agreement with the theme of efficient behaviors arising from simple rules. However, quantifying the fish's overall energy consumption in a context where they often quit is difficult: energy consumption should therefore be revisited, incorporating work on the decay of the prey capture algorithm in the last three pursuit bouts of aborted sequences (e.g. *Henriques et al., 2019*; *Johnson et al., 2019*).

With respect to hunting schemes, predatory animals have evolved a variety of strategies to optimize pursuit and intercept prey. Tiger beetles, for example, engage in pure pursuit where the angle of attack is kept constant at zero degrees (*Haselsteiner et al., 2014*). Salamanders, on the other hand, lead the trajectory of their prey (*Borghuis and Leonardo, 2015*). Dragonflies and falcons often utilize a strategy of maintaining a constant line of sight which affords the benefit of motion camouflage (*Kane and Zamani, 2014*, *Mizutani et al., 2003*; but see *Mischiati et al., 2015*). Relevantly, dragonflies also implement an implicit predictive model of their prey as well as a model of the effects of their own body movements on prey drift, which foreshadowed the possible use of

predictive models across animals with small brains (*Mischiati et al., 2015*). Larval zebrafish have been assumed to engage in pure pursuit, the simplest and most heuristic of these strategies. However, we find that the strategy used by these animals is more complex and reflects an implicit predictive model of where prey will be at a specified time in the future. Furthermore, the quantal nature of the zebrafish's swim bouts allowed us to uncover that the angle of attack is recursively and stochastically reduced by an average proportion until the prey enters a terminal strike zone.

### Embodied physical knowledge

One branch of artificial intelligence research advocates against a central processing unit where relevant computation occurs in favor of a distributed network of sensorimotor transformations, tuned to the capabilities of the body, that can accomplish the goals of the system (*Brooks, 1991*). Our study reinforces these sentiments and suggests that approaching the study of the brain without considering its embodiment may be precarious.

Specifically, interesting relationships in the data we provide suggest that the fish's algorithms are built around the capabilities and constraints of its body. First, the amount of 'fixed noise' in the fish's azimuth transformations is 7.64°. This is the standard deviation of Post-Bout Prey Az given a Pre-Bout Prey Az coordinate of 0°, the minimum of graded stochasticity observed in *Figure 6A*. The standard deviation of the strike zone itself is 7.2° in azimuth. We contend that the similarity between these two numbers suggests that the fish's strike zone is constructed to deal with noise that the fish cannot overcome in its motor program. If the prey is at, for example, 5° azimuth, performing another bout to get to the very center of the strike zone (~0° Az) would in many cases worsen the Post-Bout Prey Az coordinate due to fixed noise. Perfection is the enemy of good in this case. In this sense, the command to end pursuit and issue a strike is triggered by a visual releasing stimulus that evolved due to the fish's own bodily constraints. This is akin to the idea of embodied cognition (*Maturana and Varela, 1987*). Further evidence for embodied knowledge comes from the bias in the fish's responses to prey altitude. The fish's algorithm for transforming prey altitude biases the prey to ~18° above the fish, which aligns almost perfectly with the mean altitude coordinate at which they strike (17.4°, *Figures 2*, *3* and *5*). The mechanics of the fish jaw necessitate this: to open its jaw widely for paramecium entrance into the mouth, the fish *must* tilt its head up due to torsional constraints (*Mearns et al., 2019*). Therefore, the fish's entire sensation of prey altitude and its method of keeping the prey above it by biasing its bouts downward emerge from the way its jaw co-operates with the rest of its head. Also of note is that graded variance is minimal for altitude transformations at Pre-Bout Alt = 20° (*Figure 6A*), whereas the azimuth minimum is at 0°; this is also likely a function of its jaw features, which allows the least motor noise when the prey are located in the ideal strike position.

Finally, it is tempting to speculate that, in addition to possessing an implicit model of how objects move, the zebrafish is also equipped with a second forward model that predicts how its own body movements should give rise to expected sensory input (*Figure 5*). One way to determine the existence of this model is to test whether fish can adjust the gain of their movements in settings where they do not consistently achieve their preferred post-bout outcomes (i.e. *Ahrens et al., 2012*).

All things considered, the implicit predictive model of 3D prey motion shown in this study is: 1) embodied by the fish's stochastic recursive algorithm 2) shaped by the constraints and capabilities of the fish and 3) formulated by the interaction of three simple rules. These rules transform position of prey into fish movements, bias the vigor of fish movements based on prey velocity, and inject proportional noise into each sensorimotor transformation. Importantly, these more nuanced features of the fish's hunting algorithm would not have been revealed without examining prey capture in its more naturalistic 3D setting, which we believe has laid a groundwork for future studies examining the ontogeny, plasticity, and neural implementation of prey capture algorithms and physical knowledge in general (e.g. *Avitan et al., 2017*).

## Materials and methods

**Key resources table**

*Continued on next page*

*Continued*

| Reagent type (species) or resource | Designation | Source or reference | Identifiers | Additional information |
|---|---|---|---|---|
| Reagent type (species) or resource | Designation | Source or reference | Identifiers | Additional information |
| Software, algorithm | BayesDB | arxiv | arxiv:1512.05006 | http://probcomp.csail.mit.edu/software/bayesdb/ |
| Strain, strain background (*Danio rerio*) | *WIK* | *ZFIN* | ZFIN_ZDB-GENO-010531–2 | |

## Animals

Experiments were conducted according to the guidelines of the National Institutes of Health and were approved by the Standing Committee on the Use of Animals in Research of Harvard University. All experiments were performed on dpf 7–8 larval zebrafish of the WIK strain. Fish were raised in an automated system where they were delivered paramecia twice a day from dpf4 onward. Importantly, fish in the system experienced a full range of paramecium movement due to the height of the water in their home tank (~6'). Fish were fasted for 4–6 hr before experiments.

## Behavioral setup

After the fasting period, fish were added with ~100 paramecia (*Paramecia Multimicronucleatum*) in the dark to a 2 cm x 2 cm x 2 cm cube tank made of clear acrylic capped with coverglass. 3.56 mega-pixel images were simultaneously acquired from the top and side of the tank using two Point Grey Grasshopper 3 NIR cameras; the cameras were synchronized by a TTL pulse triggered by a Pyboard microcontroller at 62.5 Hz. Custom acquisition code was written using C# with the EmguCV library for high-speed video-writing.

Camera positions were calibrated by using known reference points (i.e. body features of the fish) for the shared plane of the cameras. Identification of known object positions in both planes was extremely accurate (~200 micron mean error), calculated by average position of the fish's eye center in both cameras over all experiments. This allowed accurate reconstruction of 3D prey and fish features (see below).

For the duration of the experiment, fish were illuminated with an infrared LED array, and after 2 min in the dark, fish were exposed to a uniform white LED which commenced prey capture. Fish hunted in the white light illuminated condition for 8 min before the experiment was terminated. Fish that did not consume more than one paramecium over the 8 min experiment were discarded for analysis (46/53 fish passed this criteria).

## Behavioral analysis

Custom Python software using the OpenCV library was written to extract the body features of the fish (eye convergence angle, tail curvature, yaw, 3D position) and the position of each paramecium in the XY and XZ planes. Pitch was calculated by taking the 2D vertical angle in the side camera and fitting a cone to the fish using the yaw angle from the top camera. The tail angle of the fish was fed to a bout detection algorithm that returned frames where swim bouts were initiated and terminated using tail angle variance and bout velocity.

Hunt sequences were identified by spectral clustering (scikit-learn) the continuous eye angle of both eyes over each swim bout for all fish into five clusters. One of the five clusters showed clear convergence of both eyes at bout initiation ('hunt initiation cluster'), while a second cluster showed clear deconvergence ('hunt termination cluster', see *Figure 1—figure supplement 1A*). Custom annotation software cycled through frames marked as hunt initiations and allowed the user to terminate hunt sequences on frames where the fish consumed its prey or clearly quit hunting. Most hunt terminations coincided with the hunt termination cluster.

Upon identifying the frame boundaries of hunt sequences, a 3D prey trajectory reconstruction algorithm was applied that matched prey discovered in the two separate cameras. This is a nontrivial task because the two cameras only overlap in one axis; any two prey items that share similar values in the overlapping plane must be separated using dynamics in time. We therefore matched prey

trajectories from the top and side using correlation of velocity profiles and post-hoc 3D position similarity. In this way, each prey item is assigned an ID for a given hunt sequence; the user is required to specify the prey ID that is struck at on strikes, and the 'best guess' prey ID that fish pursue during abort sequences. All prey trajectories for each hunt sequence are mapped to a spherical coordinate system based on unit vectors fit from the fish's XYZ position, pitch, and yaw, with its origin at the fish's oral cavity. Manual quality control for mistakes in fish characterization or prey reconstruction was applied rarely by eliminating hunt sequences from analysis showing clear mistakes from the computer vision algorithms.

## Regression fitting and modeling environment

All regression models were fit with Generalized Linear Model tools using the Python StatsModels package. When regression fits are shown in figures, we used the Seaborn library in Python, and 95% CIs for fits are represented as light shaded regions behind the regression line.

Ideal choice models used in the Virtual Prey Capture Simulation Environment cycled through bouts combined in a 'bout pool' from all 46 zebrafish that were performed during sequences that ended in a strike. Each bout during Choice was pre-filtered for bout duration before scoring for prey closeness to the strike zone; ideal bouts could not be shorter than the bout chosen by the fish at that juncture, and could not extend past the time of the next bout chosen by the fish.

Prey trajectories used in the modeling environment were selected from real capture sequences where the fish struck at the prey and the prey was swimming (>330 microns per second; 89% of all hunted prey records pass velocity criteria, which through inspection distinguishes swimming from floating prey). All virtual hunt reconstructions were initiated with the virtual fish and prey items in the exact same positions and orientations as when the real sequences were initiated. Energy consumption in the virtual environment was calculated under the assumption that the head to center of mass distance for a larval zebrafish is .53 mm (as measured in ImageJ) and the mass of the fish is 1 mg (*Avella et al., 2012*). Rotational energy of yaw and pitch and kinetic energy of center of mass displacement were added for each bout. Strike zone achievement was defined by the 95% CI on the angular position of a prey item during successful strikes (*Figure 2B*), likewise conditioned on the radial distance being less than two standard deviations from the mean.

Both regression and ideal models choose initiation bouts and pursuit bouts independently. The first bout of regression models is fit on only initiation bouts, and the first bout of choice models is chosen from the pool of all initiation plus pursuit bouts. This is largely because initiation bout transformations are significantly different from pursuits (*Figure 5—figure supplement 3*).

## Abstracted models and Bayesian nonparametric methods

Pseudocode describing the transformations of pre-bout to post-bout paramecium locations (Appendix, *Figure 5D*) were written according to the method of *Russell and Norvig (2010)*. The Appendix contains all pseudocode required to implement the deterministic and stochastic choices made in *Figures 5* and *6*.

We inferred mixture models in *Figure 6* from empirical data (Pre-Bout Prey Az, Alt, Dist and Post-Bout Prey Az Alt, Dist for all pursuit bouts in the dataset) using a non-parametric Bayesian prior called a Dirichlet Process Mixture Model (DPMM) (*Rasmussen, 1999*; *Antoniak, 1974*; *Mansinghka et al., 2016*). In order to accurately reflect realistic, stochastic pre-bout to post-bout transformations, our model choice had to be multivariate, heteroskedastic, and include multi-modal probability distributions over pursuit choices. While our linear parametric models (*Figure 5A*) captured the average transformation made by the fish in multiple velocity conditions, analytically tractable model families are unable to qualitatively capture the above phenomena. DPMMs can approximate a broad class of multivariate distributions without requiring a priori specification of the number of components in the mixture model. The mixture models generated via a DPMM prior can be converted to probabilistic programs for inference to generate the kinds of conditional simulations used in *Figure 6* (*Saad et al., 2019*). In this representation, each pre-bout to post-bout prey transformation made by a zebrafish can be thought of as arising from a program that first chooses a prototypical transform (corresponding to a component in the mixture), and then generates a random transform from a distribution over transforms associated with the prototype. We used the BayesDB software library (*Mansinghka et al., 2015*; *Saad and Mansinghka, 2016*) to implement the

computations needed to build these models and generate conditional simulations. BayesDB simulations were embedded inside a recursive loop that take an initial prey position as input and output the number of bouts until striking (see PREYCAPTURE algorithm in *Figure 5D* with STOCHASTIC_-TRANSFORM substitution). When comparing deterministic and stochastic models in *Figure 6*, the initiation bout for both models was equal and deterministic; only pursuit bouts differed between deterministic and DPMM models. The deterministic model transformed using the average slopes of 10,000 samples generated from the DPMMs to isolate stochastic effects. For validation of noise injection in *Figure 6C and D*, the GRADED_VARIANCE algorithm in the Appendix was used.

## Data and software availability

All software related to behavioral analysis, modeling, and virtual prey capture simulation is freely available at www.github.com/larrylegend33/PreycapMaster (copy archived at https://github.com/elifesciences-publications/PreycapMaster; *Bolton, 2019*). The software is licensed under a GNU General Public License 3.0. Source data for analysis and simulations is enclosed as 'Source Data' in relevant figures. Source Data for *Figure 2* contains all pursuit bouts analyzed in the dataset; it was used to construct *Figures 2*, *3*, *5* and *6A*, and is accompanied by instructions for running queries. Source Data for *Figure 6* contains the generators for simulating from the DPMMs in *Figure 6*. Using the code at www.github.com/larrylegend33/PreycapMaster and the generators in Source Data – *Figure 6* requires obtaining the BayesDB software package, which is freely available at http://prob-comp.csail.mit.edu/software/bayesdb/.

## Acknowledgements

The authors thank Martha Constantine-Paton, Mehmet Fatih Yanik, Misha Ahrens, Rory Kirchner, Rob Johnson, Lilach Avitan, Roy Harpaz, Kirsten Bolton, and Elizabeth Spelke for conversations on the project. Yarden Katz, Olivia McGinnis, and Hanna Zwaka provided helpful advice on the manuscript. Armin Bahl and Kristian Herrera provided advice and assistance with 3D rendering. This work was funded by a U19 grant from the National Institutes of Health.

## Additional information

### Funding

| Funder | Grant reference number | Author |
|---|---|---|
| National Institutes of Health | U19NS104653 | Florian Engert |

The funders had no role in study design, data collection and interpretation, or the decision to submit the work for publication.

### Author contributions

Andrew D Bolton, Conceptualization, Data curation, Software, Formal analysis, Investigation, Methodology; Martin Haesemeyer, Josua Jordi, Conceptualization, Methodology; Ulrich Schaechtle, Software, Methodology; Feras A Saad, Software; Vikash K Mansinghka, Conceptualization, Software, Supervision; Joshua B Tenenbaum, Conceptualization, Supervision; Florian Engert, Conceptualization, Supervision, Funding acquisition, Project administration

### Author ORCIDs

Andrew D Bolton https://orcid.org/0000-0003-3059-7226
Martin Haesemeyer http://orcid.org/0000-0003-2704-3601

### Ethics

Animal experimentation: Experiments were conducted according to the guidelines of the National Institutes of Health and were approved by the Standing Committee on the Use of Animals in Research of Harvard University. Animals were handled according IACUC protocol #2729.

Decision letter and Author response
Decision letter https://doi.org/10.7554/eLife.51975.sa1
Author response https://doi.org/10.7554/eLife.51975.sa2

## Additional files

### Supplementary files
• Transparent reporting form

### Data availability

All software related to behavioral analysis, modeling, and virtual prey capture simulation is freely available at https://github.com/larrylegend33/PreycapMaster (copy archived at https://github.com/elifesciences-publications/PreycapMaster). The software is licensed under a GNU General Public License 3.0. Source data for analysis and simulations is enclosed as "Source Data" in relevant figures. Source Data for Figure 2 contains all pursuit bouts analyzed in the dataset; it was used to construct Figures 2, 3, 5, and 6A, and is accompanied by instructions for running queries. Source Data for Figure 6 contains the generators for simulating from the DPMMs in Figure 6. Using the code at https://github.com/larrylegend33/PreycapMaster and the generators in Source Data - Figure 6 requires obtaining the BayesDB software package, which is freely available at http://probcomp.csail.mit.edu/software/bayesdb/.

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

# Appendix 1

```
function STRIKE(prey_coordinate) returns true if prey in strike zone and false otherwise
        inputs: prey_coordinate, a percept of the current prey position
        local variables: strikezone, 95% CI of strike probability based on 2B norm fits, or bounds of a
fixed window for single coordinate
            if prey_coordinate in strikezone then return true
            else return false
```

```
function ALT_COEFFS(prey_alt) returns alt_coefficients, a list containing a slope and
y-intercept for deterministic transform of alt coordinates
        inputs: prey_alt, a percept of the current altitude of the prey item
        local variables: alt_slope | prey_alt_positive = .54
                        alt_yint | prey_alt_positive = 8.34°                        from
                        alt_slope | prey_alt_negative = .92                        Figure 5B
                        alt_yint | prey_alt_negative = 7.03 °

        if prey_alt > 0, then return [alt_slope | prey_alt_positive, alt_yint | prey_-
alt_positive]
        else return [alt_slope | prey_alt_negative, alt_yint | prey_alt_negative]
```

```
function DETERMINISTIC_TRANSFORM(prey_coordinate) returns new_prey_coordinate,
        inputs: prey_coordinate, a percept of the current spherical prey coordinate as a
list ['az', 'alt', 'dist']
        local variables: az_slope = .53
                        alt_slope
                        alt_yint  from Figure 5A                        from
                        dist_slope = .84                        Figure 5A
                        dist_yint = -.0125 mm
                        new_prey_coordinate, the new spherical prey position after
transform
        alt_slope, alt_yint ←ALT_COEFFS(prey_position['alt'])
        new_prey_coordinate  ←[prey_coordinate['az'] * az_slope,
                        prey_coordinate['alt'] * alt_slope + alt_yint,
                        prey_coordinate['dist'] * dist_slope + dist_yint]
        return new_prey_coordinate
```

```
function PREYCAPTURE(prey_coordinate, bout_counter) returns
bout_counter, the number of bouts required for capture
        inputs: prey_coordinate, a percept of the current prey in
spherical coordinates
                bout_counter,the number of bouts the agent
has performed since hunt initiation

        if STRIKE(prey_coordinate):
                bout_counter  ←bout_counter + 1
                return bout_counter
        else:

  prey_coordinate ← DETERMINISTIC_TRANSFORM(prey_posi-          ** replace DETERMINISTIC_TRANS-
tion)                                                           FORM(prey_position)
                bout_counter ← bout_counter + 1                  with STOCHASTIC_TRANSFORM
                PREYCAPTURE(prey_coordinate, bout_counter)       (prey_position) to sample DPMM
                                                                 (see Figure 6), which implements
                                                                 graded variance
```

```
function GRADED_VARIANCE(prey_coordinate, bout_counter, dist_or_angle) returns bout_counter, the
number of bouts required for capture
        inputs: prey_coordinate, a percept of the current prey position in spherical coordinates
                bout_counter, the number of bouts the agent has performed since hunt initiation
                dist_or_angle, string representing whether input is a distance or an azimuth angle
        local_variables:
                μ a value representing the average transform
                σ the standard deviation of the average transform that decreases with proximity to the strike
zone
        if STRIKE(prey_coordinate):
                bout_counter ← bout_counter + 1
                return bout_counter
        else:
                if dist_or_angle == 'angle':
                    μ←53* prey_coordinate
                    σ ← .36 * prey_coordinate + 7.62°
                if dist_or_angle == 'distance':
                    μ←84* prey_coordinate - .0125 mm
                    σ ← 0.137 * prey_coordinate + 0.034 mm
                prey_coordinate ← GAUSSIAN_DRAW(μ, σ)
                bout_counter ← bout_counter + 1
                GRADED_VARIANCE(prey_coordinate, bout_counter, dist_or_ang)
```

