## [Decision Letter]

**Acceptance summary:**

This paper presents a highly detailed account of the kinematics of larval zebrafish hunting paramecia and compares these to various computational models to infer key aspects of zebrafish pursuit strategy and control. The work employs a free behavioral assay in which zebrafish can pursue their prey in 3D space, and takes advantage of the distinct pursuit "bouts" that zebrafish engage in during prey pursuits, unlike many other predators that display continuous pursuits. The most novel and exciting aspect of this work is the finding that the stochastic ("noisy") pursuit kinematics displayed by live zebrafish actually perform better than "ideal" alternatives that are modeled, in terms of striking a beneficial balance between capture success, speed, and energetic expenditure. The authors conclude their study with a well-written Discussion that touches on many potential causes and implications of this beneficial, stochastic pursuit strategy in 3D. This work will likely be of interest to researchers interested in prey capture writ large, as well as those interested in the role of noise and strategy in animal behavior, and points towards novel models to understand goal-directed behavior in animals.

**Decision letter after peer review:**

Thank you for submitting your article "Elements of a stochastic 3D prediction engine in larval zebrafish prey capture" for consideration by *eLife*. Your article has been reviewed by three peer reviewers, and the evaluation has been overseen by a Reviewing Editor and Ronald Calabrese as the Senior Editor. The following individuals involved in the review of your submission have agreed to reveal their identity: Phillip Washbourne (Reviewer #1); Stacey A Combes (Reviewer #2).

The reviewers have discussed the reviews with one another and the Reviewing Editor has drafted this decision to help you prepare a revised submission.

As the reviews were largely positive on the technical aspects of the work, most of the necessary revisions detailed below relate to how the work is presented. Most broadly, there was a consensus amongst the reviewers that the writing was somewhat unclear and jargon-full at times, making it less accessible to the broad *eLife* readership than it could be, so the manuscript could greatly benefit from additional copy editing and tightening (including potentially making some figures supplementary). The manuscript also would greatly benefit from additional exposition explaining the technical choices made (including, for example, the DPMM) and the consequences of these choices over other possibilities.

All of that being said, all reviewers expressed excitement about the scientific results presented in the work, and we look forward to seeing the revised submission.

Essential revisions:

- As the reviews were largely positive on the technical aspects of the work, most of the necessary revisions detailed below relate to how the work is presented. Most broadly, there was a consensus amongst the reviewers that the writing was somewhat unclear and jargon-full at times, making it less accessible to the broad *eLife* readership than it could be, so the manuscript could greatly benefit from additional copy editing and tightening (including potentially making some additional figures supplementary).

- The manuscript also would greatly benefit from additional exposition explaining the technical choices made (including, for example, the DPMM) and the consequences of these choices over other possibilities.

- The finding that larval zebrafish use both the position and velocity of their prey to predict future prey locations has been shown in many other animals, including invertebrates such as dragonflies. The reviewers didn't find this result particularly surprising, although there was less familiarity with the zebrafish prey capture literature amongst the reviewers than with other studies on other predator-prey systems – so perhaps this overturns some long-standing assumptions about the capabilities of larval zebrafish. If this is the case, the authors should emphasize more strongly why these findings are surprising. Otherwise, the findings concerning the benefit of "noisy" pursuit trajectories should receive more of the emphasis in the Abstract, conclusions, etc.

---

## [Author Response]

Essential revisions:- As the reviews were largely positive on the technical aspects of the work, most of the necessary revisions detailed below relate to how the work is presented. Most broadly, there was a consensus amongst the reviewers that the writing was somewhat unclear and jargon-full at times, making it less accessible to the broad eLife readership than it could be, so the manuscript could greatly benefit from additional copy editing and tightening (including potentially making some additional figures supplementary).

As you’ll see we have quite extensively tightened the text and we paid particular attention in focusing on the main message and on de-jargonizing. We think it’s much, much better now and more readily accessible to a broad audience.

- The manuscript also would greatly benefit from additional exposition explaining the technical choices made (including, for example, the DPMM) and the consequences of these choices over other possibilities.

We have added to the Results section where we give a more detailed explanation for our choices of models; and we also moved the more technical aspects to the Materials and methods.

- The finding that larval zebrafish use both the position and velocity of their prey to predict future prey locations has been shown in many other animals, including invertebrates such as dragonflies. The reviewers didn't find this result particularly surprising, although there was less familiarity with the zebrafish prey capture literature amongst the reviewers than with other studies on other predator-prey systems – so perhaps this overturns some long-standing assumptions about the capabilities of larval zebrafish. If this is the case, the authors should emphasize more strongly why these findings are surprising. Otherwise, the findings concerning the benefit of "noisy" pursuit trajectories should receive more of the emphasis in the Abstract, conclusions, etc.

We thank the reviewers for this helpful suggestion. Indeed, we do not find this result particularly surprising since a lot of animals do use combined velocity and position estimations to guide their behavior. We now explicitly cite and discuss work from the dragonfly and salamander literature that addresses this point. However, we note that this question was unresolved in larval zebrafish and it is therefore clearly important and interesting (if not surprising) that larval zebrafish do this, too. Also, velocity perception is fundamental to our recursive algorithm and is as such an essential component. We have modified manuscript to make this clearer and we also shifted the emphasis onto the noisy pursuit strategy, citing additional biological stochasticity papers in the Introduction, emphasizing the stochasticity in the Abstract, and adding a section on its novelty in the Discussion.